# Ossifying Fibroma in the Nasal Cavity of a 2-Year-Old Horse

**DOI:** 10.3390/ani11020317

**Published:** 2021-01-27

**Authors:** Bernard Turek, Kamil Górski, Olga Drewnowska, Roma Buczkowska, Natalia Kozłowska, Rafał Sapierzyński

**Affiliations:** 1Department of Large Animals Diseases and Clinic, Institute of Veterinary Medicine, Warsaw University of Life Sciences (SGGW), Nowoursynowska 159, 02-776 Warsaw, Poland; kamykg@o2.pl (K.G.); olga_drewnowska@sggw.edu.pl (O.D.); roma_buczkowska@sggw.edu.pl (R.B.); 2Department of Pathology and Veterinary Diagnostic, Institute of Veterinary Medicine, Warsaw University of Life Sciences (SGGW), Nowoursynowska 159, 02-776 Warsaw, Poland; rafal_sapierzynski@sggw.edu.pl

**Keywords:** tumor, ossifying fibroma, tooth, extraction, computed tomography

## Abstract

**Simple Summary:**

This article reports on the clinical features, diagnosis, and management of ossifying fibroma in the nasal cavity of a 2-year-old horse. Ossifying fibromas (OFs) are rare, benign, fibro-osseous neoplasms that occur more frequently in the rostral mandible of young horses (termed equine juvenile mandibular ossifying fibromas) but rarely in older horses. The occurrence of OF in young animals suggests developmental disorders or trauma as etiological factors. The local recurrence of OFs is common if they are not completely surgically excised, but metastatic spread is unusual. The presented case remained clinically asymptomatic until the mass obliterated the whole nasal cavity, causing severe breathing difficulties. The exact mass location was revealed using diagnostic images—namely, radiographs and computed tomography (CT) images. A concurrent problem of an underdeveloped and hypoplastic last premolar tooth of the maxilla was diagnosed. Because the mass was well-demarcated, the horse underwent standing surgery to remove the mass and the tooth. Histopathological diagnosis of ossifying fibroma was confirmed. The patient recovered uneventfully and remained free of disease at the 2-year postoperative follow-up.

**Abstract:**

A 2-year-old mare of an unknown breed was referred to the clinic due to undetermined breathing difficulties. Physical examination revealed painless swelling rostral to the nasoincisive notch and a large, firm mass protruding from the left nostril. Radiographic examination of the head revealed a mass occupying the left nasal cavity and a displaced and hypoplastic last premolar of the left maxilla. The CT scan showed a well-demarcated heterogeneous mass measuring 22 × 9 × 5 cm (length × height × width) in the left nasal cavity attached to the roots of the displaced tooth and conchae. The surgery was performed on the standing horse. Firstly, due to the oblique position of the displaced tooth, the extraction was performed extra-orally through the trephination and repulsion of the maxillary bone. In the next step, a direct surgical approach was chosen for the caudal part of the mass via the osteotomy of the left nasal bone. The mass was bluntly separated from the conchae and removed through the nostril using Fergusson forceps. The histopathological characteristics of the mass led to the diagnosis of ossifying fibroma. The horse recovered completely in seven months, without recurrence after two years.

## 1. Introduction

There are no clear-cut data on the incidence of tumors in the skeletal system in horses. Only in one paper was the rate of occurrence reported to be below 1.5% [1].

Ossifying fibromas (OFs) are fibro-osseous neoplasms that most commonly occur on the intramembranous bone of the skull and mandible [2] and rarely in the appendicular skeleton [3]. Fibro-osseous lesions have been reported in humans [4,5] and a variety of domestic animals, including dogs [6], cats [7], rabbits [8], llamas [9], birds [10], and ruminants [11,12,13].

In horses, ossifying fibromas usually affect the rostral part of the mandible of young animals aged between 2 and 15 months and are termed equine juvenile mandibular ossifying fibromas (EJMOFs). These lesions may replace alveolar and cortical bone, cause the loosening of the teeth, interfere with the mastication of food, and predispose the jaw to pathological fractures [14]. Although the cause of OF is unknown, several hypotheses have been suggested, including developmental disorders that may affect ossification at a young age as well as trauma [15]. Other reports have described OF in horse maxilla [16], paranasal and sphenopalatine sinuses [17,18,19], proximal tibia [20], and the fourth metacarpal bone [21].

Ossifying fibromas tend to recur if they have not been completely removed, but metastatic spread is unusual [22]. The objective of the present paper is to describe the clinical, imaging, and pathological findings of an ossifying fibroma in the nasal cavity of a horse.

## 2. Materials and Methods

### 2.1. Case Presentation

A 2-year-old mare of an unknown breed was admitted to the clinic due to breathing difficulties. On initial examination, the general conditions of the patient were assessed as poor because of malnutrition, poor quality of fur, and general neglect. Breathing was stridulous, with no passage of air through the left nostril. Temperature and pulse were within normal limits. A deformation was visible on the left side of the head, rostral to the nasoincisive notch. Epiphora was noted in the left eye. During a detailed examination, a firm, non-shifting mass protruding from the left nostril was found (Figure 1).

### 2.2. X-ray, CT, Endoscopy Examination

Radiographs were obtained by computed radiography (Gierth HF 80 Plus, Gierth X-ray International, Germany, and Console Advance DR-ID 300 CL, Fujifilm Corporation, Tokyo, Japan) with an indirect conversion system for scanning. The left latero-lateral (LL) and right 30° dorsal–left ventral oblique (Rt30D-LeVO) projections were assessed. A well-delineated, oval-shaped, and slightly heterogeneous mineral opacity was seen within the left nasal passage, extending from the level of 208 to the nostril (Figure 2a) [23]. Triadan 208 was displaced into the rostral aspect of the nasal cavity. The morphology of tooth 208 was abnormal, with markedly variable radiopacity and hypoplastic reserve, and clinical crown. Deciduous teeth 607 and 608 were visible (Figure 2b). The right and left maxillary sinuses remained intact.

The endoscopic examination of the right nostril revealed the rightward bending of the nasal septum; an examination of the left side was unavailable due to the presence of the mass. Computed tomography of the head was performed under general anesthesia in dorsal recumbency. The patient was sedated with detomidine (0.02 mg/kg) prior to anesthetic induction with diazepam (0.1 mg/kg) and ketamine (2.2 mg/kg). Anesthesia was maintained using isoflurane (induction 5%, maintenance 2–3%). CT images were acquired with a Revolution 750 CT scanner, GE Healthcare Corporation, Chicago, IL, USA (120 kV, 280 mA, at 1.3 mm slice thickness). Two-dimensional tomographic sections were viewed using the RadiAnt DICOM 2020.2.3 software. Multiplanar image reconstructions allowed for the identification of the exact location of the mass, which extended from the level of Triadan 208 to the rostral part of the nasal cavity. The space-occupying lesion was well defined, heterogeneous, and measured approximately 22 × 9 × 5 cm (length × height × width) with internal zones of calcification (Figure 3). The mass caused the deformation and compression of the left dorsal and ventral nasal conchae, the deviation of the nasal septum, and the thickening of the right dorsal and ventral nasal conchae (Figure 4a). The tooth was displaced palatally to the nasal cavity, and the morphology was abnormal due to hypoplastic reserve, clinical crown and roots, enlarged infundibula, and variable radiodensity. The apex of 208 was deformed and had a direct connection to the mass (Figure 4b). The sinuses appeared normal.

### 2.3. Surgery

Surgery was performed using the following steps: extraction of the deciduous teeth intra-orally, extraction of Triadan 208, and removal of the tumor. All procedures were performed in a standing position during the same anesthetic protocol.

The horse was sedated using detomidine (0.02 mg/kg IV) and butorphanol (0.01 mg/kg IV), maintained with constant infusion rates for detomidine (0.02 mg/kg/h) and butorphanol (0.01 mg/kg/h) diluted in saline, delivered by an infusion pump. The dose was adjusted according to the horse’s behavior. Maxillary cheek teeth were anesthetized with a left infraorbital nerve block using 5 mL 2% lidocaine hydrochloride [24]. The left maxillary nerve was blocked caudally from the maxillary foramen with 20 mL of 2% lidocaine hydrochloride [25]. The horse was monitored with a vital sign monitor (ECG, pulse oximetry, capnography). The deciduous teeth 607 and 608 were removed intra-orally without any complications. Due to medial displacement and the oblique position of the Triadan 208, it was impossible to grasp the crown with the molar forceps. An extra-oral technique was used that involved an approach to the roots through the left maxillary bone trephination and extraction by repulsion [26]. A 6 cm incision was made through the skin, and subcutaneous tissue levator nasolabialis and caninus muscles were bluntly separated. The periosteum was elevated and pushed off the bone. A 25 mm diameter trephine was used for creating a hole in the bone rostral and dorsal facial crest, below the infraorbital foramen. A disc of bone was removed to expose the roots. A dental punch of 10 mm size was used for the repulsion of the tooth into the mouth. During the surgery, the crown of the diseased tooth was palpated orally to detect vibrations and confirm the correct positioning of the dental punch. The tooth was subsequently removed in one piece using dental elevators. At the end of the surgery, a drain was placed in the wound. It was removed 3 days after surgery, and no complications in healing were observed. Periosteum and subcutaneous tissue were closed using a simple continuous suture pattern (absorbable monofilament thread 1/0), and the skin was apposed with 8 widely spaced, simple-interrupted, non-absorbable sutures (monofilament thread 1/0).

Extensive damage to the apex of the alveolus created an oronasal fistula measuring approximately 2 cm in length by 1.5 cm in width. In the initial treatment, lavage and debridement were performed. In the next step, surgical gauze with iodoform was immersed in a semi-liquid paste of plaster of Paris then compressed slightly and placed in the fistula [27].

To remove the mass, it was necessary to make an additional dorsal approach through the osteotomy of the nasal bone to release the caudal connection between the mass and nasal conchae [28]. After the incision of the skin (10 cm), extending to the subcutaneous tissues, blunt dissection was used to displace the levator nasolabialis and levator labii maxillaris muscles. The periosteum was elevated. An oscillating saw was used to create a rectangular bone fragment measuring 2 × 5 cm caudally to the nasoincisive notch (Figure 5). The removal of the bone allowed direct access to and inspection of the caudal part of the nasal mass, which had a pedunculated attachment and was adjacent to the dorsal and ventral conchae. The connection was removed with a periosteum separator by blunt dissection. Minimal bleeding occurred. The mass was separated and extracted through the nostril using Fergusson forceps (Figure 6).

The skin and subcutaneous tissues were sutured as previously described (Figure 7). Part of the mass was sent for histopathological examination.

### 2.4. Postoperative Treatment

After surgery, the horse received an administration of procaine penicillin (80,000 IU/kg IM) and streptomycin (8 mg/kg IM), which was continued postoperatively for 10 days. During the first week, flunixin meglumine (1.1 mg/kg) was administered intravenously as anti-inflammatory therapy. After maxillary tooth extraction, the fistula was lavaged with physiological saline solution, debrided, and plugged with iodoform gauze immersed in plaster of Paris to allow granulation and to prevent the persistence of an oronasal communication. Three, four, and nine days later, under sedation (detomidine 0.02 mg/kg IV and butorphanol 0.01 mg/kg IV) the plug was replaced and the fistula was lavaged and debrided. Sixteen days after initial surgery, a polyvinyl siloxane plug (PVS) was placed in the alveolus. The PVS plug was changed once a week under sedation, as previously described. To improve the healing process, during each procedure fistula irritation with surgical gauze and a miniature tube brush was performed, and a new dental impression material plug was molded and inserted. Eight weeks later, endoscopic examination revealed that the oral opening of the fistula was filled with granulation tissue and mucosa, but the persistence of a 2 mm-diameter oronasal communication in the apical region of the alveolus was detected.

The apical part of the plug was remodeled and adjusted to the alveolus. The plug was frequently loosened, necessitating its replacement and resulting in the contamination of the nasal cavity with feed. In the 12th week of treatment, the plug was modified by adding a small plastic sleeve into the PVS packing material, to which a nylon thread was attached. Free ends were fastened to a button fixed on the surface of the maxillary bone (Figure 8). Additional injections of the fistula with platelet-rich plasma (PRP) were conducted in a once-weekly regimen for 3 weeks. Endoscopic examination performed in the 16th week of treatment revealed a markedly reduced tract of the fistula to a 1 mm diameter (Figure 9). Two months later, an autologous cancellous bone graft was taken from tuber coxae to promote bony fusion and revascularization. First, the oronasal fistula was filled with graft and the alveolar cavity was then covered with a modified polyvinyl siloxane plug, as previously described, for 4 weeks. Seven months after the initial extraction, endoscopic examination revealed the complete closure of the fistula.

## 3. Results

The results of the treatment should be considered satisfactory, despite the long-lasting complications associated with the presence of a connection between the nasal and oral cavity. The fistula healed and, consequently, closed within 7 months after the surgical procedure. Histopathology revealed the benign proliferation of fibrous connective tissue superficially (Figure 10a), with the presence of irregularly shaped trabeculae of woven bone rimmed with osteoblasts separated by fibrous stroma embedded in the fibrous tissue. In the fibrous area, proliferating fibroblast-like spindle cells, collagen fibers, and blood vessels were present. Only a few mitotic figures were observed (Figure 10b). The histopathologic findings were consistent with ossifying fibroma.

To date—i.e., 2 years after surgery—no recurrence was observed, and the horse does not show any disturbing symptoms.

## 4. Discussion

The exact etiology of ossifying fibroma is unknown. Most OFs are found in young animals and have been reported to be present already at birth [29]. In humans, OFs are presumed to originate from mesenchymal blast cells in the periodontal ligament, not as a neoplasm but as areas of excessive bone resorption with attempted repair [30]. Some authors believe that such tumors are not neoplastic but developmental in origin, while others consider them as the ossification of changes associated with dysplasia of fibrous tissue [31]. These theories on the origin of OF remain inconclusive. In the case presented here, there was a concurrent problem with a hypoplastic and displaced tooth that had a connection with the mass. The mobility and root reabsorption of the teeth involved have frequently been described in the literature [32]. We suspect that the tumor originated from the periodontal ligament of Triadan 208, was the result of excessive tooth repair, or that the tooth problems appeared after expansile mass growth.

The most frequent location of OF is mandibular, involving the premolar and molar region [33]. OF in animals has no clear sex predilection, but in humans it has typically been more prevalent in females than males [34]. It is manifested by slow-growing progressive and painless mass formation [35]. The neighboring tissues can be pushed without being destroyed [36]. Signs of early mass growth are unspecific, causing delay in diagnosis. The case presented here was clinically asymptomatic until the tumor occupied the whole nasal cavity, causing severe breathing difficulties. A presumptive diagnosis was based on clinical examination, endoscopy, radiography, and computed tomography evaluation. In differential diagnosis, a cyst; inflammatory lesion; benign tumor, such as osteoma; fibrous dysplasia; and a tumor with a predilection for paranasal sinuses, including squamous cell carcinoma, undifferentiated carcinoma, adenocarcinoma, and osteosarcoma, were taken into consideration [22,37,38]. On radiographs, the extent of the mass could be assessed, but the exact location and the involvement of the different compartments of the nasal cavity could not be verified.

Computed tomography (CT) overcame the limits of radiography, allowing for the good visualization of the size, localization, and impact on surrounded structures; it also provided the veterinary surgeon clear indications for the planned surgery [39,40]. However, the exact differentiation of the soft tissue structures was not possible. The deformation and compression of the left dorsal and ventral nasal concha were visible on the images, but full assessment could be achieved only during surgery. The CT appearance of ossifying fibromas may vary widely depending on the degree of ossification [39]. It has been observed that the edges of the lesion are usually well defined. The internal structure shows a mixed radiodensity, with a pattern that depends on the form and quantity of the calcified material [41]. The only disadvantage of a CT scan is the need for general anesthesia, which entails considerable risk of complications during both the anesthetic period and recovery [42]. A definitive diagnosis was based on histopathology [43]. The differentiation of certain benign neoplasms such as osteoma and fibrous dysplasia may be difficult [44]. It has been suggested that ossifying fibroma has the features of both osteoma and fibrous dysplasia [18]. One report described a case of osteoma with features of an ossifying fibroma [45]. There is also a suggestion that an OF may mature into an osteoma [29].

The horse’s temperament and behavior allowed the surgery to be conducted as a standing procedure; therefore, the cost of the procedure and the risk of postoperative complications of standing up due to hemorrhage were significantly reduced. Surgery started with the dental extraction of Triadan 208. In this case, a dorsal approach to the roots through the external maxillary bone trephination and repulsion was recommended and more appropriate than the lateral approach through the buccal soft tissues (buccotomy) because of the oblique position and displacement into the nasal cavity [46].

The persistence of oro–nasal fistula was a problematic complication. Probably in the initial period of treatment, the polyvinyl siloxane plug was incorporated too high, which prevented the alveolar closure of the upper part by granulation tissue. A later problem was keeping the packing material firmly in the alveolus. It was necessary to modify this method by adding a small plastic sleeve into the PVS packing material, which was attached by nylon thread to the button fixed on the surface of the maxillary bone. This method allowed the alveolar material to be maintained for a long time. The only observed side effect was that skin inflammation occurred due to the sutured button.

To release the mass completely, it was necessary to make an additional excision of a nasal bone osteotomy. A direct approach to the nasal cavity via osteotomy over the nasal bone was described and is recommended for conditions such as an ethmoid hematoma, paranasal sinus cyst, and neoplasia, or in cases in which the preoperative diagnosis is uncertain [47,48,49]. This approach allows the tumor to be easily and quickly released from the surrounding tissues. The wound was traditionally closed with a simple interrupted suture, which provided good conditions for wound revision if necessary.

OFs tend to recur if they are not completely excised, and follow-up radiography is advised to monitor any re-growth [2,38]. As an alternative, radiation and chemotherapy have been used effectively to treat the recurrence of these tumors after surgery or as sole treatment [38,50]. Furthermore, complete surgical excision is associated with a higher number of successful reported cases [2,4,14,30].

## 5. Conclusions

In conclusion, signs of early mass growth are unspecific and cause a delay in diagnosis. A preliminary diagnosis of ossifying fibroma can be made from its clinical appearance, its site of development, the age of onset, radiography, and CT, with subsequent definitive diagnosis after histopathology. Total removal is considered ideal for OF and is reported to reduce recurrence and improve the outcome.

In this paper, computed tomography played an important role in the diagnostic procedure. Surgery was successful because the mass was well circumstanced. A direct approach to the nasal cavity through a bone appeared to be a good technique for tumor removal from the nasal cavity. In the presented case, the extraction and removal of the tumor was achieved, though the oronasal fistula represented the most challenging problem. A satisfying outcome was achieved using a modified PVS plug, but a significant improvement in complete closure was noticeable after the application of an autologous bone graft.

## Figures and Tables

**Figure 1 animals-11-00317-f001:**
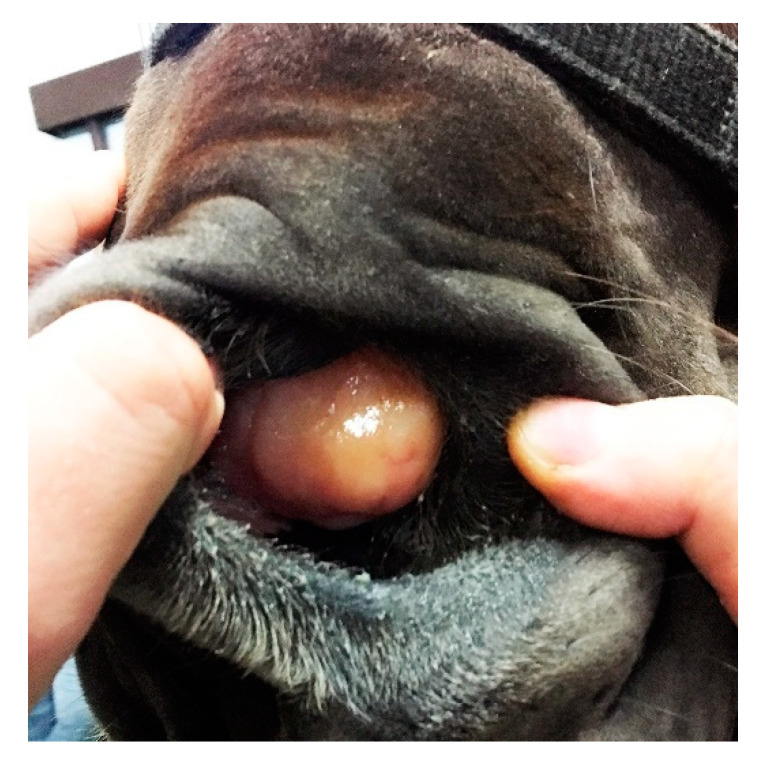
Firm mass protruding from the left nasal cavity of a mare with breathing difficulties.

**Figure 2 animals-11-00317-f002:**
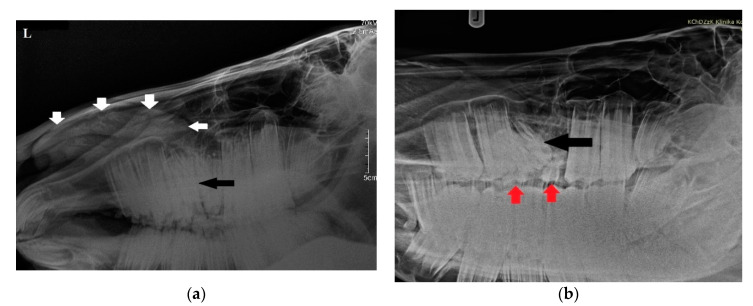
(**a**) Radiographic examination in latero-lateral projection demonstrating a well-demarcated mass within the nasal cavity (white arrows) and displaced Triadan 208 (black arrow). (**b**) Rt30D-LeVO projection of the maxillary region revealing hypoplastic dislocated maxillary last premolar (black arrow). Clearly visible deciduous Triadan 607 and 608 (red arrows).

**Figure 3 animals-11-00317-f003:**
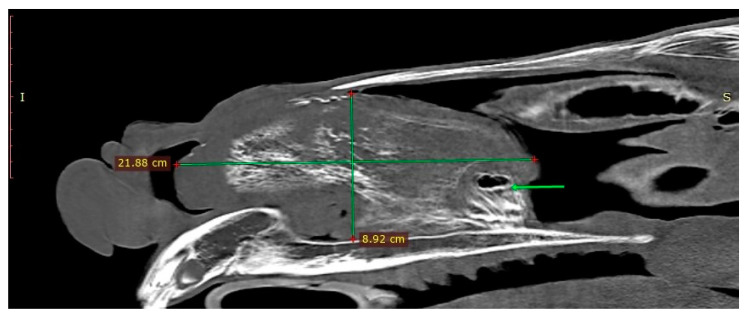
Sagittal CT image obtained left to sagittal line (bone algorithm, WW1500, WL 300). The mass length × height are indicated with green lines. Connection between the mass and the Triadan 208 is present (green arrow).

**Figure 4 animals-11-00317-f004:**
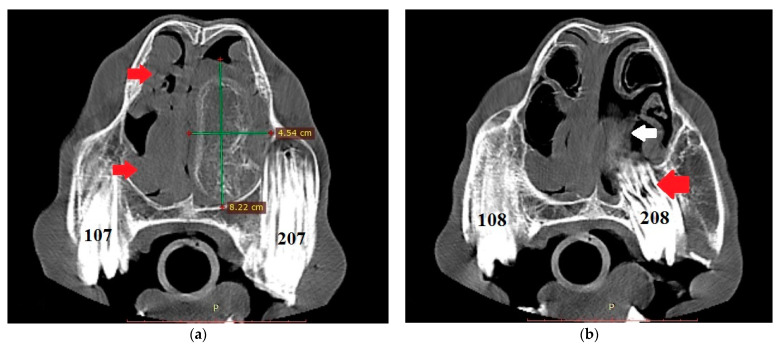
(**a**) Transverse CT image at the level of Triadan 07 (bone algorithm, WW1500, WL 300), demonstrating a heterogeneous mass obliterating the left nasal cavity (width × height—green lines). Note the rightward deviation of the nasal septum and thickened right dorsal and ventral nasal conchae (red arrows). Internal zones of calcification are present. (**b**) CT transverse plane at the level of Triadan 08 (bone algorithm, WW1500, WL 300). Note the relationship of the mass (white arrow) with the dental apex of Triadan 08 (red arrow).

**Figure 5 animals-11-00317-f005:**
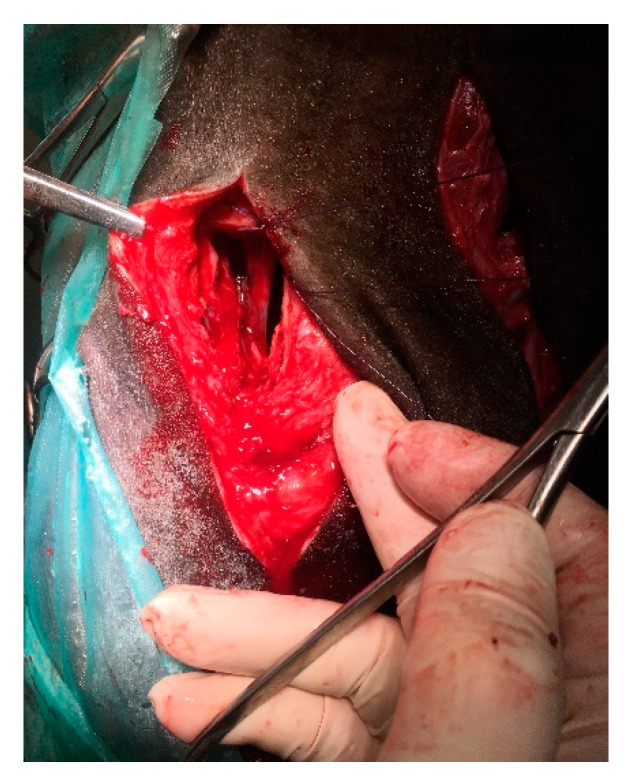
Intraoperative view. Dorsal approach to the caudal part of the tumor by the osteotomy of a 2 × 5 cm bone fragment dorsally to the infraorbital foramen.

**Figure 6 animals-11-00317-f006:**
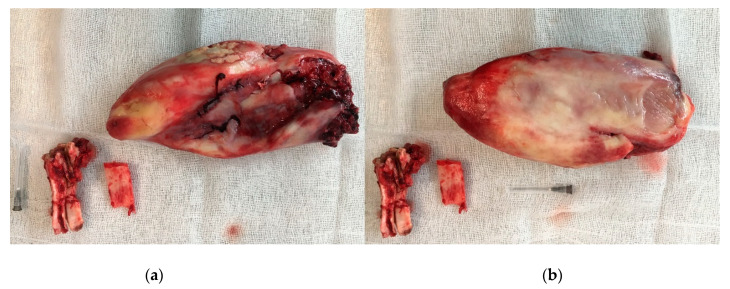
Extracted Triadan 208, fragment of the nasal bone, excised mass: (**a**) lateral site, (**b**) medial site.

**Figure 7 animals-11-00317-f007:**
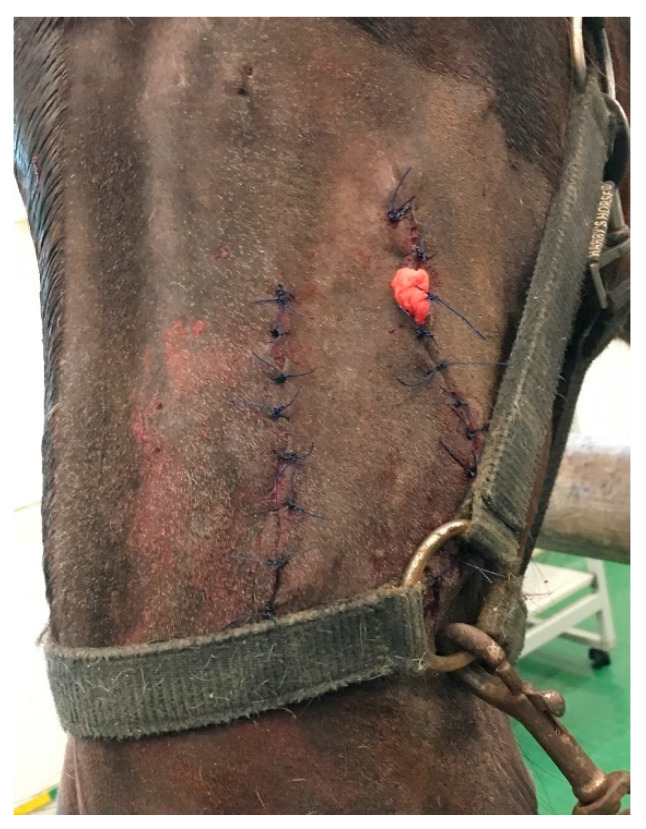
Postoperative view. Both incisions were closed with a simple interrupted suture. A drain was left in the wound after tooth extraction. It was removed 3 days after surgery, and no complications in healing were observed.

**Figure 8 animals-11-00317-f008:**
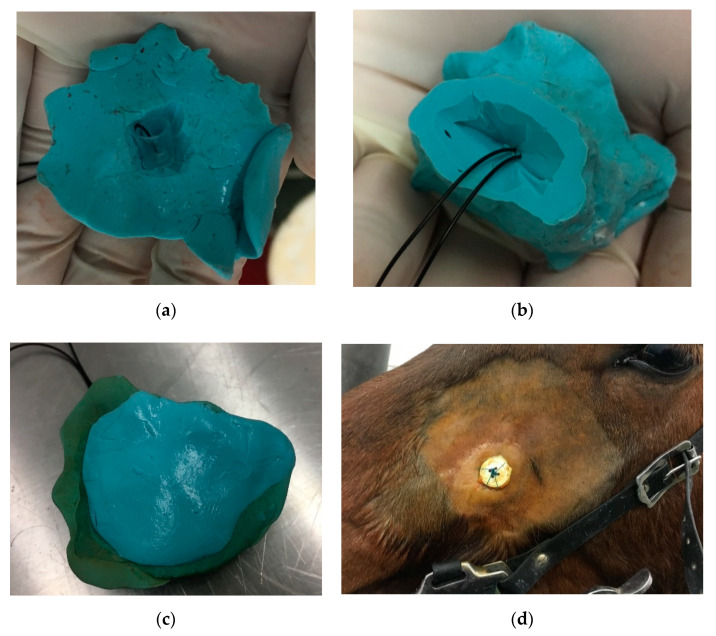
Polyvinyl siloxane impression material inserted into newly created oro-nasal fistula fastened with thread and button: (**a**) occlusal surface, (**b**) alveolar surface, (**c**) sealed occlusal surface, (**d**) button at the skin surface.

**Figure 9 animals-11-00317-f009:**
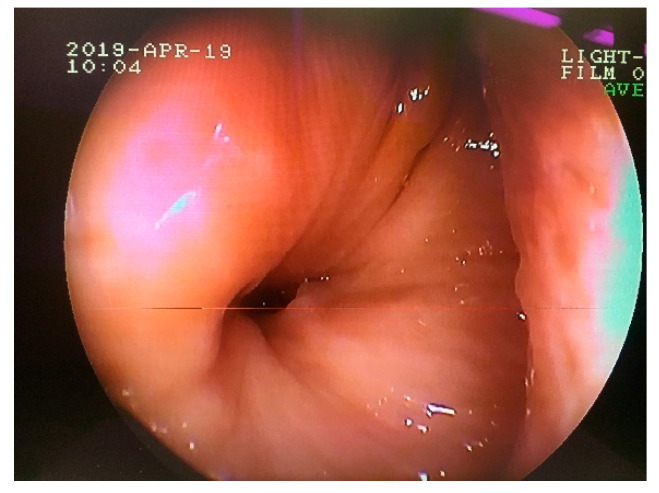
Transnasal endoscopic examination obtained 16 weeks after the initial surgery. Note the nasal opening of the persisted oronasal fistula at the level of 208 alveolus.

**Figure 10 animals-11-00317-f010:**
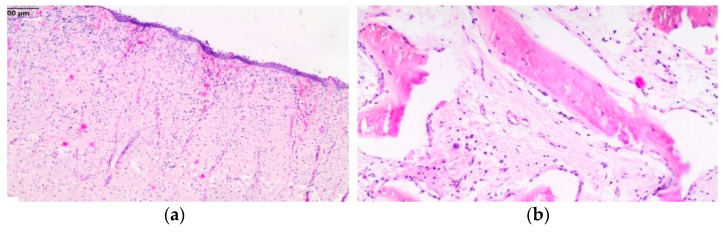
Histological section of the mass. Benign proliferation of fibrous connective tissue superficially (**a**); curvilinear trabeculae of woven bone set in a fibrous stroma in deeper parts of the mass (**b**).

## Data Availability

Not applicable.

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
