# Peer review of "Ossifying Fibroma in the Nasal Cavity of a 2-Year-Old Horse"

_animals, 2021, doi:10.3390/ani11020317_

Round 1

Reviewer 1 Report

General comments:

  • English needs to be improved
  • Need to use Triadan system nomenclature
  • Need to specify at the begging the abbreviations, such as CT
  • Title, need to specify that this is a case report of a single case
  • The term tumor is consistently used thoroughly in the manuscript for the description of the radiographic and CT images. However, this term is not correct from the descriptive point of view and should be replaced by a synonym.
  • Very poor quality of the images.

Simple summary:

Line 21: use radiography instead of x-ray and computed tomography instead of CT

Abstract

Line 30: when describing radiographic findings the term tumor is not adequate as there are other differential diagnoses to be considered

Line 32: if you describe the size of a structure using a tomographic imaging modality you need to provide the 3 dimensions and not only 2

M&M

Imaging:

It is necessary to provide the details of the equipment used.

The radiographic description is short and lacks descriptive terms. Firstly, the term tumor is not correct in a description. A mass or space-occupying lesion can be used instead. More information about the exact location of the mass, the approximate size would be beneficial.

Need to specify the radiographic protocol.

Is it possible to describe the 208 as dysplastic based on the radiographic appearance?

If the dose of the drugs used during induction is provided, the dose of isoflurane used would be also necessary.

In a CT examination only transverse images are obtained, however, in the post-processing multiplanar reconstruction can be performed, so the sentence in line 85 is not correct.

The description of the CT images is confusing, as its exact location is not stated. What do you mean by the connection of the mass with the last premolar?

What is the relationship of this mass with the nasal septum and the nasal conchae?

There is no description of the teeth.

Was there any sinus involvement?

Surgery:

The surgical approach and procedures are confusing and not clearly described, especially the first step.

Results:

The histopathological examination and the outcome of the horse should be included in this part.

Figures:

Figure 2:

It seems that the figures are photographs of the screen, the quality is very poor and non-diagnostic. Need to provide original files.

Images are not labeled.

For image B it is necessary to provide the correct name of the projection in the legend.

Figure 3:

It seems that the figure is a photograph of the screen, the quality is very poor and non-diagnostic. Need to provide the original file.

The legend says that the figure is at the level of 207, but this figure is approximately at the level of the interdental space.

What do you mean with remodeling of the adjacent structures?

What do you mean with facial bones? This term is not correct from an anatomical point of view.

The use of arrows would be beneficial.

Figure 4:

It seems that the figure is a photograph of the screen, the quality is very poor and non-diagnostic. Need to provide the original file.

Longitudinal is incorrect, it should be sagittal.

The use of arrows would be beneficial.

What do the yellow lines show?

Reviewer 2 Report

Line 14:  of an ossifying

Line 16:   , and rarely

Line 21: using radiographs and a CT scan.

Line 22: Because the  mass was well-demarcated, the horse underwent standing surgery to remove the tooth and the tumor. The surgical procedure was performed in two stages.

Line 24: confirm an…

Line 27: of undetermined

Line 30: and a displaced

Line 31: The CT scan showed

Line 32: as a two stage procedure

Line 34: through an osteotomy of the maxillary bone…. The second stage consisted on creating a nasal bone flap that allowed

Line 42: recovered completely in seven months without recurrence two years later

Line 52: affect

Figure 2 b: the image looks like an screen picture. Do you have the original radiographic image? As it is the case with A

Figure 3: same problem, is there a better quality image?

Figure 4: the mass was

Line 103: Tooth extraction

Line 107: to the horse behavior

Line 111-112: phrase is not clear. Maxillary bone flap?

Line 113-117: paragraph is not clear. Was the bone flap closed? The dental alveoli was damaged during surgery? Or was food there already?

Line 119: through an

Line 120: just a “nasal bone flap” or was part of the nasal bone removed?

Lines 120-123: why is figure 6 presented before figure 5? Verify numeration

Line 125: separated and extracted

Line 133: including

There are two Figure 6 !!!

Line 146-147: how was the bone graft fixed to the fistula? How was food prevented from passing through? Special diet? How many bone grafts were required? It took 7 months, how many procedures?

Lines 153-158 repetitive from introduction

Lines 179-180: the risk of postoperative risk (please correct)

Figure 7: this explains part of the procedures done for handling the fistula. They should not be presented on the discussion, and a more detail explanation of how this fistula was treated would be worth. Is the picture D from the same case?

All the description of this is in the discussion, it should be placed before

Figure 10: the format is not correct, and B is not explained in the figure legend

Line 230: growth

Line 237: in the case presented here…

Round 2

Reviewer 1 Report

The manuscript has been significantly improved after the first review, but still there are some parts that need to be improved.

English still needs to be implemented

Line 98: details of the radiographic equipment used are not provided

Lines 98-100. Use the correct nomenclature consistently in the entire manuscript: left latero-lateral (LL) and right 30ºdorsal-left ventral oblique (Rt30D-LeVO) projections

Smallwood, J., Shively, M., Rendano, V., & Habel, R. (1985). A standardized nomenclature for radiographic projections used in veterinary medicine. Veterinary Radiology, 26(1), 2–9.

Line 100 It is better to make a general radiographic description rather than describing each projection separately. Radiographs do not show anything, lesions can be seen or identified on the radiogrpahs. Describing the mass using the terms radiodense and mineral opacity can be redundant. The description of the location of the mass can be improved as difficult to understand as is stated now.

A more precise description would be: a well-delineated, oval-shaped and slightly heterogeneous mineral opacity is seen within the left nasal passage, extending from the nostril to the level of 208.

Line 103: which crown do you refer to: reserve o clinical?

Line 104: which maxillary sinus?

Line 124: use a dot instead a coma when describing decimals (0.1 and not 0,1). This needs to be changed in several parts of the manuscript

Line 130: Which dimensions refers to each plane? A common way to describe measurements is: 1x1x1 cm (RCdxDVxML)

Line 133: change medially to palatally, turbinates or conchae?

Line 134: which crown are you refering to?; Change infundibulum to infundibula

Figure 3: flip the image horizontally (rostral facing the left side)

Nasal septum?

Change high to height

Figure 4:

High to height

Can you show the distortion of the maxillary bones?

What do you mean with enlargement of the right sided nasal conchae?

Line 151: I cannot see the relationship of the mass with the tooth on the images, what the arrows show can also represent a partial volume artifact. Could you provide an image where this is clearly seen?

Line 180: Dorsal and not above

Line 190: Need to be consistent with the description of the measurements, if you describe in brackets the length you also need to the same with the width

Line 203: NASAL conchae (review the manuscript for using the same term)

Line 204: All the anatomical references has been provided in English, so the name of the muscles should follow the same rule.

Figure 6: Use the Triadan system (still there are parts of the manuscript where this is not used)

Figure 7: Incisions and not wounds

Description of the use of a drainage should be in the text and not in the figure legend

Line 233: When starting a sentence using a number it should be written with letters: Three, 3 and 9

Line 236, 252: Same as before
